# Endoscopic and Image Analysis of the Airway in Patients with Mucopolysaccharidosis Type IVA

**DOI:** 10.3390/jpm13030494

**Published:** 2023-03-09

**Authors:** Yi-Hao Lee, Chin-Hui Su, Che-Yi Lin, Hsiang-Yu Lin, Shuan-Pei Lin, Chih-Kuang Chuang, Kuo-Sheng Lee

**Affiliations:** 1Department of Otorhinolaryngology-Head and Neck Surgery, Hsinchu Branch of MacKay Memorial Hospital, Hsinchu 30071, Taiwan; 2Department of Otorhinolaryngology-Head and Neck Surgery, Hsinchu Municipal MacKay Children’s Hospital, Hsinchu 30070, Taiwan; 3Department of Otorhinolaryngology-Head and Neck Surgery, MacKay Memorial Hospital, Taipei 10449, Taiwan; 4School of Medicine, Taipei Medical University, Taipei 11031, Taiwan; 5Department of Otorhinolaryngology-Head and Neck Surgery, National Taiwan University Hospital, Taipei 10002, Taiwan; 6Department of Medicine, MacKay Medical College, New Taipei City 25245, Taiwan; 7Department of Pediatrics, MacKay Memorial Hospital, Taipei 10449, Taiwan; 8Department of Medical Research, MacKay Memorial Hospital, Taipei 10449, Taiwan; 9The Rare Disease Center, MacKay Memorial Hospital, Taipei 10449, Taiwan; 10Nursing and Management, MacKay Junior College of Medicine, Taipei 11260, Taiwan; 11Department of Medical Research, China Medical University Hospital, China Medical University, Taichung 40402, Taiwan; 12Department of Infant and Child Care, National Taipei University of Nursing and Health Sciences, Taipei 11219, Taiwan; 13College of Medicine, Fu-Jen Catholic University, Taipei 24205, Taiwan; 14Department of Audiology and Speech-Language Pathology, MacKay Medical College, New Taipei City 25245, Taiwan

**Keywords:** airway obstruction, tracheal stenosis, glycosaminoglycans (GAGs), mucopolysaccharidosis (MPS), Morquio syndrome, fiberoptic bronchoscopy, computed tomography, T-tube

## Abstract

Mucopolysaccharidosis (MPS) is a hereditary disorder arising from lysosomal enzymes deficiency, with glycosaminoglycans (GAGs) storage in connective tissues and bones, which may compromise the airway. This retrospective study evaluated patients with MPS type IVA with airway obstruction detected via endoscopy and imaging modalities and the effects of surgical interventions based on symptoms. The data of 15 MPS type IVA patients (10 males, 5 females, mean age 17.8 years) were reviewed in detail. Fiberoptic bronchoscopy (FB) was used to distinguish adenotonsillar hypertrophy, prolapsed soft palate, secondary laryngomalacia, vocal cord granulation, cricoid thickness, tracheal stenosis, shape of tracheal lumen, nodular deposition, tracheal kinking, tracheomalacia with rigid tracheal wall, and bronchial collapse. Computed tomography (CT) helped to measure the deformed sternal angle, the cross-sectional area of the trachea, and its narrowest/widest ratio (NW ratio), while angiography with 3D reconstruction delineated tracheal torsion, kinking, or framework damage and external vascular compression of the trachea. The NW ratio correlated negatively with age (*p* < 0.01), showing that airway obstruction progressed gradually. Various types of airway surgery were performed to correct the respiratory dysfunction. MPS type IVA challenges the management of multifactorial airway obstruction. Preoperative airway evaluation with both FB and CT is strongly suggested to assess both intraluminal and extraluminal factors causing airway obstruction.

## 1. Introduction

Mucopolysaccharidosis (MPS) is a hereditary disorder caused by the deficiency of lysosomal enzymes. This deficiency results in the storage of glycosaminoglycans (GAGs) in the connective tissues throughout the body, particularly in the bones, as well as in the central and peripheral nervous system, liver, blood vessels, skin, cartilage, airways, heart valves, and cornea [1]. The current classification categorizes MPS disorders into seven types based on clinical features and enzyme deficiencies: type I H/S/H-S (Hurler/Scheie/Hurler–Scheie syndrome), type II (Hunter syndrome), type III A/B/C/D (Sanfilippo syndrome), type IV A/B (Morquio syndrome), type VI (Maroteaux–Lamy syndrome), type VII (Sly syndrome), type IX (Natowicz syndrome). MPS type IVA (Morquio syndrome type A) is characterized as a deficiency of N-acetylglucosamine-6-sulfate sulfatase (GALNS) enzyme, resulting in excess accumulation of GAGs, such as chondroitin-6-sulfate (C6S) and keratan sulfate (KS) [2,3].

The most common otorhinolaryngologic manifestations in the majority of MPS patients include adenotonsillar hypertrophy, macroglossia, recurrent otitis media with effusion, and hearing loss [4,5,6,7,8]. Pharyngeal obstruction by soft tissue not only increases the difficulty of endotracheal intubation, but also leads to snoring and obstructive sleep apnea. Accumulation of GAGs in the laryngopharyngeal soft tissue and tracheobronchial cartilage may cause both upper and lower airway obstruction and respiratory dysfunction, especially in MPS types I, II, IVA, VI, and VII [6,7,8,9,10,11,12].

Patients with MPS type IVA usually have skeletal dysplasia, presenting with short stature, genu valgum, pectus carinatum, kyphosis, abnormal gait, and laxity of the wrist joints [13]. Severe bony deformity may also compromise the airway. The intrathoracic structures may be crowded as a result of a deformed thoracic cadge and narrowed thoracic inlet, which may lead to tracheal compression by external vessels or soft tissue [3,12,14,15]. Disproportionate growth between chest cavity, neck, and trachea (tracheal overgrowth) may also lead to tracheal kinking and obstruction [3,12,14,15]. Instability of the cervical spine (C1–2) and spinal cord compression may also compromise the respiratory function due to dysfunction of the phrenic nerve and inspiratory muscle weakness [16,17].

As GAGs accumulate and chronic disease progresses with age, patients with MPS type IVA typically have a reduced lifespan, with mortality between the ages of 20 and 30 years, and mortality is mostly associated with airway obstruction or spinal cord compression [3,12,16]. Early intervention and treatment for the obstructive airway disease can prolong the lifespan of patients with MPS IVA and improve their quality of life. However, multifactorial airway obstruction with chest deformity and instability of the cervical spine complicates the airway management in patients with MPS IVA, making it extremely challenging. Individualized treatment can be determined in advance through detailed airway examinations using different imaging tools. Both fiberoptic bronchoscopy (FB) and computed tomography (CT) were used for airway evaluation in the present study. These advanced imaging tools allowed the observation of the different characteristics of airway obstruction and the measurement of the severity of stenosis. The study purpose was to evaluate both the upper and the lower airway in patients with MPS type IVA through endoscopy and imaging modalities and to investigate the effects of surgical interventions on symptoms.

## 2. Materials and Methods

### 2.1. Study Design and Sample

This single-center retrospective study collected the data of 38 cases of MPS type IVA from the hospital records of MacKay Memorial Hospital, Taipei, Taiwan, over 20 years, from 2001 to 2021. Inclusion criteria were: clinic visits in the ear–nose–throat (ENT) department or being referred by pediatricians for the evaluation of obstructive airway, including examinations of the airway from the nasopharynx to the main bronchi performed by FB or CT. Patients whose medical records were incomplete or whose images were not uploaded into the Picture Archiving and Communication System (PACS) were excluded. Patients who had visited the ENT clinic or been referred from other departments, but without examination of the airway, were also excluded. Finally, 15 patients were enrolled in this study.

### 2.2. Ethical Considerations

The study protocol was approved by the MacKay Memorial Hospital Institutional Review Board (IRB number: 22MMHIS324e). Because of the retrospective design of this study and the deidentification of patient data, the IRB waived the signed informed consent of the included patients.

### 2.3. Fiberoptic Bronchoscopy (FB)

Physical examination of the pharynx is used to grade tonsillar hypertrophy and macroglossia. However, deeper structures are usually evaluated via FB, which can be performed under local anesthesia, passing through the nasal cavity, nasopharynx, oropharynx, hypopharynx, larynx, cricoid, trachea, carina, and bilateral main bronchi to examine both upper and lower airways. FB is used to evaluate (1) adenoid hypertrophy, (2) tonsillar hypertrophy, (3) prolapsed soft palate (high larynx), (4) secondary laryngomalacia, (5) vocal cord granulation, (6) cricoid thickness, (7) tracheal stenosis, (8) shape of the tracheal lumen, (9) deposition of nodules in the airway, (10) tracheal kinking, (11) tracheomalacia with rigid tracheal wall, and (12) bronchial collapse (Figure 1). The severity of each specific feature is defined in Table 1.

Certain characteristics of the endoscopic findings can only be recorded as “present (+)” or “absent (–)”, while adenotonsillar hypertrophy, macroglossia, secondary laryngomalacia, and tracheal stenosis can be graded by scales. Through physical examination, tonsillar size is graded from 0 to 4, and Friedman tongue position (FTP) is evaluated on a scale from I to IV, as previously described [18]. Hypertrophic adenoids, which results in nasal obstruction and obstructive sleep apnea syndrome, can also be assessed by transnasal FB and graded from 0 to 4, as previously described [19]. Laryngomalacia usually presents with redundant supraglottic tissue with or without prolapse into the laryngeal inlet, causing airway obstruction. Though it is difficult to quantify the definitive severity of tracheal stenosis by FB, the segment with most severe stenosis was recorded and graded according to the four grades of the Myer–Cotton classification [20]. In addition, various shapes of the tracheal lumen observed with FB were recorded at the levels of the upper, middle, and lower segments of the trachea.

**Table 1 jpm-13-00494-t001:** (a) Parameters of bronchoscopy and definitions of grading. (b) Parameters of computerized tomography and definitions of grading.

Parameters		Grading	Measurement
**(a)**			
Pharynx	Adenoids [19]	0	Post adenoidectomy
		1	Adenoid tissue not in contact with other structures
		2	Adenoid tissue in contact with torus tubaris
		3	Adenoid tissue in contact with torus tubaris, vomer
		4	Adenoid tissue in contact with torus tubaris, vomer, soft palate
	Tonsils [18]	0	No tonsils seen in pillars or post tonsillectomy
		1	Within the pillars
		2	Extended to the pillars
		3	Extended past the pillars
		4	Extended to the midline
	Tongue [18]	I	Full visibility of uvula, tonsils and pillars
		IIa	Visibility of most of the uvula but not of the entire tonsils/pillars.
		IIb	Visualization of the entire soft palate to the uvular base.
		III	Partial visibility of the soft palate with distal end absent
		IV	Visibility of only the hard palate
	Prolapsed soft palate (high larynx)	–	Uvula not touching the epiglottis
		+	Uvula touching the epiglottis
Larynx	2^0^ LM	–	Smooth supraglottic tissue
		+	Redundant supraglottic tissue, without prolapse or airway compromise
		++	Redundant supraglottic tissue, with prolapse and airway compromise
	VC granulation	–	Thin and smooth vocal cord
		+	Vocal cord bulging and granulated change
	Cricoid thickness	–	Cricoid cartilage smooth with patent lumen
		+	Cricoid cartilage thickening with narrowed lumen
Trachea	Stenosis [20]	I	0% to 50% decrease in lumen surface
		II	51% to 70% decrease
		III	71% to 99% decrease
		IV	No evidence of detectable lumen
	Shape	C	C-shaped lumen, normal tracheal morphology
		U	U-shaped lumen, mild to moderate lateral collapse of tracheal lumen
		D	D-shaped lumen, mild to moderate anterior–posterior collapse of tracheal lumen
		T	Triangular lumen, moderate to severe tracheal deformity caused by multiple factors
		W	Worm-shaped lumen, severe tracheal deformity caused by multiple factors
		B	Tracheal lumen in the shape of the Mercedes–Benz logo, severe tracheal deformity caused by multiple factors
	Deposit nodule	–	No evidence of GAGs deposition in the tracheal lumen
		+	Nodular deposition of GAGs in the tracheal wall
	Kinking	–	No sharp-angled kinking in the tracheal lumen
		+	Tracheal torsion and bending with sharp-angled kinking in the lumen
	TM with rigid lumen	–	Tracheoesophageal membrane stays supple
		+	Deformed tracheal ring and lumen with circumferential stiffness, involving the tracheoesophageal membrane
Bronchus	Collapse	–	Bronchial lumen patent
		+	Bronchial lumen collapsed
**(b)**			
Chest	Deformity	–	Normal thoracic cage and sternal curve
		+	Thoracic cage and sternum deformed, with angle of curved sternum >135 degrees
		++	Thoracic cage and sternum deformed, with angle of curved sternum 90~135 degrees
		+++	Thoracic cage and sternum deformed, with angle of curved sternum <90 degrees
	Sternal angle	Measuring the angle of the deformed sternum by tools of the PACS	
Trachea	Stenosis	–	Normal shape of the trachea, or NW ratio of 76–100%
		+	Mildly collapsed, NW ratio of 51–75%
		++	Moderately collapsed, NW ratio of 26–50%
		+++	Severely collapsed, NW ratio <25%
	NW ratio	Measuring the ratio of the narrowest and widest cross-sectional area by PACS	
	Torsion	–	Straight and smooth trachea
		+	Deformed trachea with torsion or bending morphology
	Kinking	–	Straight and smooth trachea
		+	Deformed trachea with sharp-angled kinking
	External compression	–	Straight and smooth trachea
		+	Deformed trachea caused by innominate artery or aortic arch external compression
	Framework damage	–	Straight and smooth trachea
		+	Deformed trachea with framework damage

2^0^ LM, secondary laryngomalacia; VC, vocal cord; GAGs, glycosaminoglycans TM, tracheomalacia; NW ratio, narrowest and widest area ratio; PACS, Picture Archiving and Communication System.

### 2.4. Computed Tomography (CT)

The clinical features were assessed and recorded as follows: (1) severity of chest deformity, (2) sternal angle (curve), (3) severity of tracheal collapse, (4) ratio of the narrowest to the widest areas (NW ratio) of the tracheal lumen, (5) tracheal torsion (6) tracheal kinking, (7) external vascular compression of the trachea, and (8) tracheal framework damage (Figure 2). The severity of each specific feature is defined in Table 1.

Using the tools of the PACS, CT was used to measure the angle of the deformed sternum and the cross-sectional area of the airway. The severity of the chest deformity was defined according to the curved sternal angle. Tracheal collapse, which is difficult to quantify precisely using only FB, was graded according to the ratio between the narrowest and the widest areas (NW ratio) of the tracheal lumen. CT angiography with three-dimensional reconstruction was also used to demonstrate the structure of the entire cartilaginous airway from the trachea to the bronchi, as well as the relationship between the external vessels and the trachea.

### 2.5. Surgical Treatment for the Airway

Airway surgery was required if obstructive sleep apnea or respiratory dysfunction was noted. The surgical treatment for each case was recorded, and a pathologic examination was performed using special histochemical stains (colloidal iron) to evaluate GAGs deposition when the specimens were obtained. 

### 2.6. Statistical Analysis

Pearson’s correlation was used to assess associations between continuous variables such as age, sternal angle, and NW ratio, since the disease is supposed to progress with age. Categorical variables were analyzed by the percentage of the subjects.

## 3. Results

### 3.1. Patients’ Baseline Characteristics

The data of 15 MPS type IVA patients (10 males, 5 females) were reviewed in detail, (Table 2). The ages on examination ranged from 7 to 30 years, with a mean age of 17.8 years (median age 17 years). Four patients had previously undergone enzyme replacement therapy (ERT) for 5 months to 10 years. The detailed findings of the imaging exams for each patient’s specific airway characteristics are listed in Table 2. The distribution of each endoscopic and image finding as well as the surgical type were also statistically analyzed, and the results are shown in Table 3.

### 3.2. Fiberoptic Bronchoscopy

#### 3.2.1. Pharynx

The adenoid size exceeded grade 2 in 80% (12/15) of the patients and was grade 2 in 60% (9/15) of them. Two patients with grade 2 were evaluated post-adenoidectomy and showed regrowth of the lymphoid tissue. Grade 3 was the most common size of the palatine tonsil and was determined in 80% (12/15) of the patients, while the remaining 3 patients were grade 0 (postoperative), 2, and 4. Macroglossia was observed through physical examination rather than endoscopy, but the characteristic was still recorded to complete the pharyngeal findings. Most patients had severe macroglossia, including 20% (3/15) of them with grade III and 73.3% (11/15) with grade IV, while only one patient was grade II. Prolapsed soft palate, also known as high larynx, was found in 80% (12/15) of the patients.

#### 3.2.2. Larynx

The main manifestations of secondary laryngomalacia included redundant supraglottic tissue and deformed arytenoid or epiglottis. About 60% (9/15) of the patients had severe secondary laryngomalacia with airway obstruction, while 26.7% (4/15) had an open laryngeal inlet despite the presence of redundant supraglottic tissue. Granulation of vocal cords or false cords caused by chronic inflammation and deposition was found in 53.3% (8/15) of the patients. In addition, thickening of the cricoid cartilage with narrowed lumen was noted in 76.9% (10/13) of the patients.

### 3.3. Trachea and Bronchi

Among all patients, 41.7% (5/12) had grade I tracheal stenosis by Myer–Cotton classification, 16.7% (2/12) had grade II, and 41.7% (5/12) had grade III, but none had progressed to grade IV. The shapes of the tracheal lumen at the upper, middle, and lower levels of the trachea were also recorded using FB. A normal C-shaped tracheal lumen was only seen in 33.3% (4/12) of the patients. U-shaped and D-shaped tracheal lumens presenting as mild to moderate tracheal stenosis were observed in 58.3% (7/12) and 25% (3/12) of the patients, respectively. T-shaped (triangular shape), W-shaped (worm shape), and B-shaped (Mercedes-Benz logo) tracheal lumens were noted in 50% (6/12), 25% (3/12), and 16.7% (2/12) of the patients and could lead to severe tracheal collapse needing further airway management. Other tracheal abnormalities were observed in two patients who showed nodular deposition of GAGs in the lower airway. Tracheal kinking with a sharp-angle tracheal lumen was also seen in two patients. Tracheomalacia with a generally rigid tracheal lumen involving the trachea–esophageal membrane was noted in 41.7% (5/12) of the patients. However, no bronchial collapse or stenosis was noted in any patient.

### 3.4. Computed Tomography

#### 3.4.1. Chest

Bony dysplasia with severe chest deformity and pectus carinatum represent distinctive characteristics associated with MPS type IVA. The mean angle of the sternum was 107.75° (ranging from 44° to 162°), defined as moderate deformity. About 76.9% (10/13) of the patients had moderate (++) to severe (+++) chest deformity, while only three had a normal to mild (+) form. The severity of the deformed chest cage and sternal angle was not significantly associated with age (*p* > 0.01) (Figure 3).

#### 3.4.2. Trachea

When the cross-sectional area of the stenotic trachea was measured by PACS and the severity of stenosis was graded according to the NW ratio, the mean NW ratio was 42% (range, from 9% to 68%), which was defined as moderate tracheal collapse by CT measurement. The NW ratio correlated negatively with age (*p* < 0.01, r = −0.708) (Figure 3). With CT angiography and three-dimensional reconstruction, the tracheal morphology and the relationship between airway and external vessels were also delineated. Tracheal torsion, tracheal kinking, vascular compression of the trachea, and tracheal framework damage were 61.5% (8/13), 23.0% (3/13), 61.5% (8/13), 23.0% (3/13), respectively.

### 3.5. Surgery with Pathological Study

Upper airway obstruction usually needs surgical intervention if progression includes severe symptoms such as sleep apnea or dyspnea. Tonsillectomy was performed in 33.3% (5/15) of the patients, and adenoidectomy was performed in 37.5% (6/15) of them. Laser-supraglottoplasty was performed in 33.3% (5/15) of the patients due to secondary laryngomalacia. Moreover, 20% (3/15) of the patients with the narrowest airways underwent tracheostomy with T-tube stenting below the vocal cord to preserve both laryngeal and tracheal patency. The results of the pathological study with colloidal iron stain showed that GAGs were deposited in the tonsils, adenoids, supraglottic tissue, cricoid, and tracheal cartilage in some patients (Figure 4).

## 4. Discussion

Bony dysplasia with chest deformity is a primary characteristic of MPS type IVA, despite that chest deformity and the curved sternal angle not being significantly associated with age in the present study. In contrast, the NW ratio correlated negatively with age, indicating that the tracheal stenosis and airway obstruction progressed gradually. Thus, we suggest that regular follow-up in the ENT clinic is needed to examine the airway and to provide airway management if necessary.

While previous studies described several types of examinations for airway evaluation in MPS patients, the present study relied on the results of CT and endoscopy for most findings in patients with MPS IVA. FB is the most widely used examination method for both upper and lower airway evaluation and is able to support preoperative decision making and postoperative follow-up [6,9,21,22,23,24,25]. FB is an office-based procedure that can be performed by an experienced otorhinolaryngologist or pulmonologist transnasally under local anesthesia, with the patient in a sitting position. Rigid ventilation bronchoscopy (VB) performed in an operating room with the patient under general anesthesia is an alternative procedure that is useful in cases with poor visualization of the airway due to the severe collapse of soft tissue or to patients’ intolerance of local anesthesia. Either FB or VB allows a clear visualization of the whole airway, including the nasopharynx, oropharynx, larynx, trachea, and primary bronchus. Most patients with MPS are expected to have GAGs deposition with hypertrophic change in pharyngeal and laryngeal soft tissue, but the pattern of tracheal deformity may be diverse based on multiple factors, including nodular deposition in the tracheal cartilage, disproportionate growth between the trachea and the chest cavity causing tracheal torsion or kinking, external compression by the innominate artery or the aortic arch, or simply framework damage due to GAGs storage. In the present study, the degree of stenosis was estimated using the Myer–Cotton classification, and the shape of the tracheal lumen was also described, which provided indications of external tracheal factors. The C-shaped tracheal ring was a relatively normal anatomical structure, while the U-shaped (lateral collapse) and D-shaped (anterior–posterior collapse) tracheal rings were associated with mild to moderate narrowing. The triangular, worm-shaped, and Mercedes-Benz-shaped tracheal lumens, which presented as moderate to severe tracheal stenosis, were usually caused by crowded intrathoracic structures or external vascular compression. Bronchial collapse was not found initially in any of these patients, but was present in follow-up visits, especially in patients with airway infection or mucosal irritation as a result of multiple surgical interventions.

Although the intraluminal characteristics can be described well with FB or VB, the external factors that cause airway collapse are not revealed sufficiently by endoscopy alone. Thus, CT angiography with 3D reconstruction served as a powerful imaging tool that not only helped to measure the precise stenotic cross-sectional area and deformed sternal angle [21,25,26,27] (Figure 2A,B), but also revealed the entire appearance of the cartilaginous airway and the relationship between the trachea and the external vessels [6,24,28]. On axial view of CT images and the 3D reconstructive mode, the shape of the tracheal lumen correlated with the endoscopic findings, and the external factors that caused such collapse could be determined (Figure 2C,D). If the trachea was obstructed simply due to redundant soft tissue or nodular deposition, balloon dilatation under ventilation bronchoscopy may be helpful. However, once tracheal stenosis is associated with external vascular compression or tracheal kinking due to the overgrowth of the trachea, balloon dilatation appears to be not effective enough to widen the airway, and further stenting procedures should be considered. 

Other investigative tools such as magnetic resonance imaging (MRI), polysomnography (PSG), and pulmonary function tests (PFT) were also suggested in previous studies for further advanced respiratory evaluation [25]. However, the supraglottis and tongue base soft tissue are better assessed with MRI scans, while CT scans provide better images of the infraglottis, trachea, bronchus, and lung [24], although evaluation of tracheal stenosis using MRI has been reported [12]. PSG reveals whether or not a patient has obstructive sleep apnea alone or combined with central apnea. Central apnea in MPS type IVA is probably associated with spinal cord compression and respiratory muscle weakness due to instability of the cervical spine [16,17]. In addition, MPS type IVA patients usually have bony dysplasia, presenting with short stature, kyphosis, scoliosis, flattening of the vertebral bodies, and deformed thoracic cage leading to limited lung expansion and restrictive lung disease, which can be evaluated by PFT [6,24,25,29]. However, the results of PSG and PFT were not reported in the present study because of fragmentary data and poorly organized protocols during earlier investigations of MPS patients. 

The current treatment options for MPS include enzyme replacement therapy (ERT) and hematopoietic stem cell transplantation (HSCT). In the present study, only four MPS type IVA patients had previously undergone ERT before airway examinations. However, ERT and HSCT seemed to lack meaningful effects for managing either airway obstruction or restrictive lung disease in MPS type IVA despite long-term treatment, likely due to their limited impact on bone and cartilage lesions [3,14]. To date, there is no proof on the ability of ERT to resolve bony or cartilaginous deformity that causes airway problems in MPS type IVA patients. We also found GAGs still deposited in the cytoplasm of chondrocytes and in the extra-cellular matrix in the tracheal cartilage in one of our patients (case No. 8) after receiving ERT for 5 months (Figure 4). Continuous positive airway pressure (CPAP) or bilevel positive airway pressure (BiPAP) may effectively improve ventilation in MPS patients but may fail to improve their quality of life [6,25,29]. Surgical intervention, however, still restores airway patency effectively and simultaneously improves the quality of life.

Airway management still remains challenging during intubation due to the presence of macroglossia, short neck, submucosal and cartilaginous deposition of GAGs, and cervical spine instability. Intubation with FB assistance [30], use of a video laryngoscope [31,32,33], or even combined tools [34] have been suggested for MPS type IVA management, not only to compensate for the poor visualization of the laryngeal inlet but also to prevent the over-extension of the neck and potential spinal cord injury.

Adenotonsillectomy is the most common surgical procedure for upper airway obstruction in MPS patients [7,8,25]. Prophylactic adenotonsillectomy may also be considered in patients with MPS type IVA before undergoing fusion surgery of the cervical spine, which may increase the difficulty of exposing the surgical field for the removal of tonsils and adenoids [4].

Tracheostomy is widely performed to secure different types of upper airway obstruction in children and adults, but certain difficulties in patients with MPS type IVA limit the use of this procedure. A short neck with a deeply buried trachea complicates the approach to the trachea, and severe pectus carinatum is a challenge as well as dangerous for inserting the tracheostomy tube. The type of tracheostomy tube needs careful consideration and discussion by surgeons based on preoperative airway examinations. Customized tracheostomy is usually needed in order to stent the distal trachea due to severe tracheomalacia and progressive GAGs deposition in the airway. Phonation and swallowing disturbances are common problems after tracheostomy.

With advances in modern medicine and surgical techniques, the lifespan of MPS patients has been prolonged gradually over recent years. Gradual GAGs deposition in the airway and tracheal stenosis may progress distally with age. The tracheal stenting procedure may offer a palliative treatment for tracheobronchomalacia and lower airway stenosis in MPS patients. Antón-Pacheco et al. used stenting treatment in one patient with MPS type IVA [35], but details of the stenting material and long-term outcomes of this case were not presented in the study report. Metallic balloon-expandable stents have been applied in MPS type II and type VI patients [36,37,38,39]. However, most patients had only short-term benefits from this type of stent, and their responses were complicated by considerable granulation formation due to mucosal inflammation and GAGs deposition [37,38]. The T-tube is a good alternative stenting material first used in patients with MPS I, II, and VI by Soni-Jaiswal et al. [40]. Our team also first applied the T-tube in patients with MPS type IVA (patients No. 8~10) to preserve phonation, swallowing, and breathing in these adult patients who had social demands [15]. Although distal progression of tracheobronchomalacia and gradual GAGs deposition are inevitable due to the natural course of the disease, the T-tube provides better long-term outcomes and prolongs the inter-procedure period when compared with metallic stents.

Surgical procedures that include tracheal vascular reconstruction are more aggressive, because they require midline sternotomy and cardiopulmonary bypass, even though they are still markedly effective [14,33]. Reimplantation of the innominate artery rightward to the proximal ascending aorta not only eliminates the external vascular compression of the airway, but also allows safe the insertion of the tracheostomy tube, avoiding complications caused by tracheal innominate fistula. Segmental resection of the trachea with end-to-end anastomosis reduces tracheal torsion or kinking, keeping the airway straighter and less subject to obstruction.

This study has several limitations, including the fact that the number of cases examined was small and the data were retrospective, which limits the generalization of our results to other populations. However, as many data as possible were collected from the medical records and imaging system, sufficient to compare cases, imaging methods, and surgical interventions. Videos were usually taken during FB but were not uploaded and preserved in the PACS due to the huge data amount. It was difficult to evaluate dynamic changes in the airway during the inhalation and exhalation phases by only jumping screenshots of endoscopic images, and many details were probably missed without reviewing the videos. Lack of experience in the early days and the lack of protocols for airway examination with this type of disease preceded the more recent development of examinations for MPS type IVA. Some patients did not complete all airway examinations due to intolerance of FB under local anesthesia, relatively patent airways without obvious obstructive symptoms, or loss of follow-up because of their emigration.

## 5. Conclusions

The management of multifactorial airway obstruction in MPS type IVA is especially challenging. Although the degree of chest deformity has no correlation with age, airway stenosis progresses with age in MPS type IVA patients and may eventually need surgical intervention. Preoperative airway evaluation with both FB and CT is strongly suggested to assess both the intraluminal and the extraluminal factors that cause airway obstruction. The correlation of FB and CT images is necessary to obtain details for making the most appropriate decisions for surgical interventions to correct airway obstruction.

## Figures and Tables

**Figure 1 jpm-13-00494-f001:**
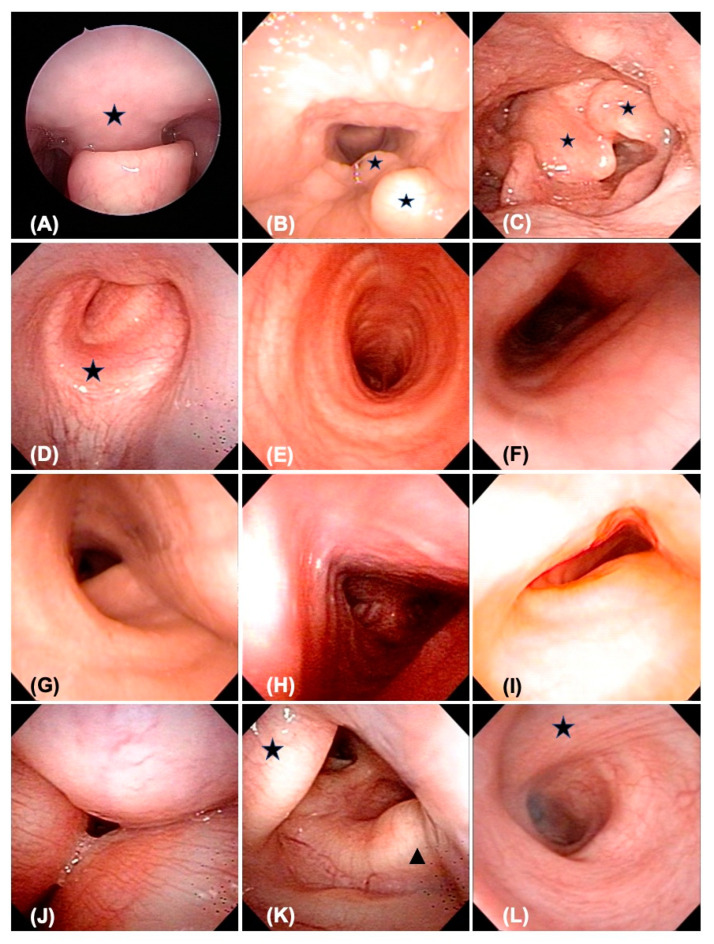
Variable characteristics of airway obstruction under fiberoptic bronchoscopy. (**A**) Uvula in the larynx with laryngeal inlet obstruction. (**B**) Granulation or GAGs deposition in vocal cord and false cord. (**C**) Secondary laryngomalacia with prolapsed supraglottic tissue. (**D**) Thickness of cricoid and narrowed lumen. (**E**) Normal C-shaped tracheal ring. (**F**) U-shaped tracheal lumen. (**G**) D-shaped tracheal lumen. (**H**) T-shaped tracheal lumen. (**I**) W-shaped tracheal lumen. (**J**) B-shaped tracheal lumen. (**K**) Tracheal kinking (star-mark) and GAGs nodular deposition (triangle-mark). (**L**) Tracheomalacia with rigid tracheal lumen involving the tracheoesophageal membrane.

**Figure 2 jpm-13-00494-f002:**
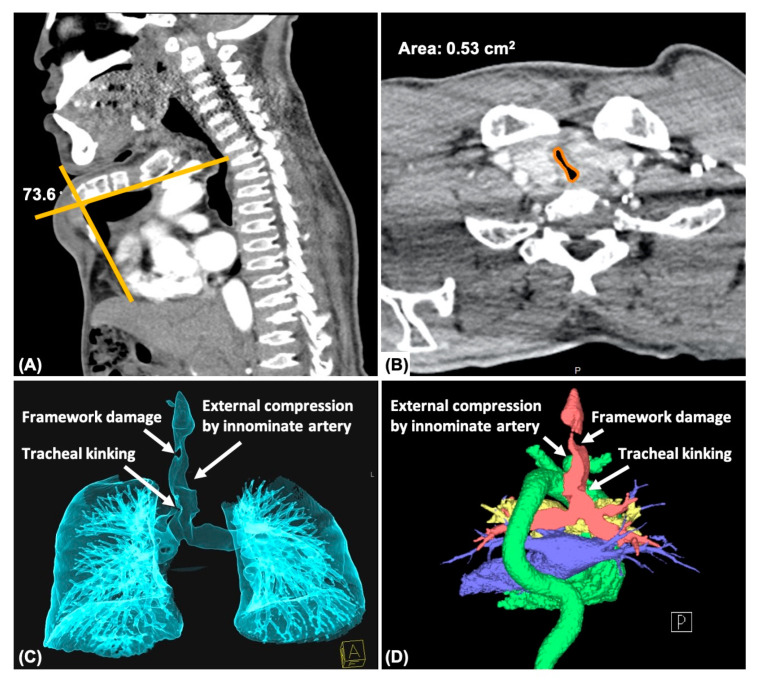
Computed tomography and angiography with three-dimensional reconstruction in patient No. 10. (**A**) Measuring the deformed sternal angle in MPS type IVA with pectus carinatum. (**B**) Measuring the cross-sectional area of the narrowed tracheal lumen and then determining the NW ratio. (**C**) Anterior view of the deformed airway. (**D**) Posterior view of the cardiovascular and tracheobronchial structures. (**C**,**D**) As described in previous studies [15].

**Figure 3 jpm-13-00494-f003:**
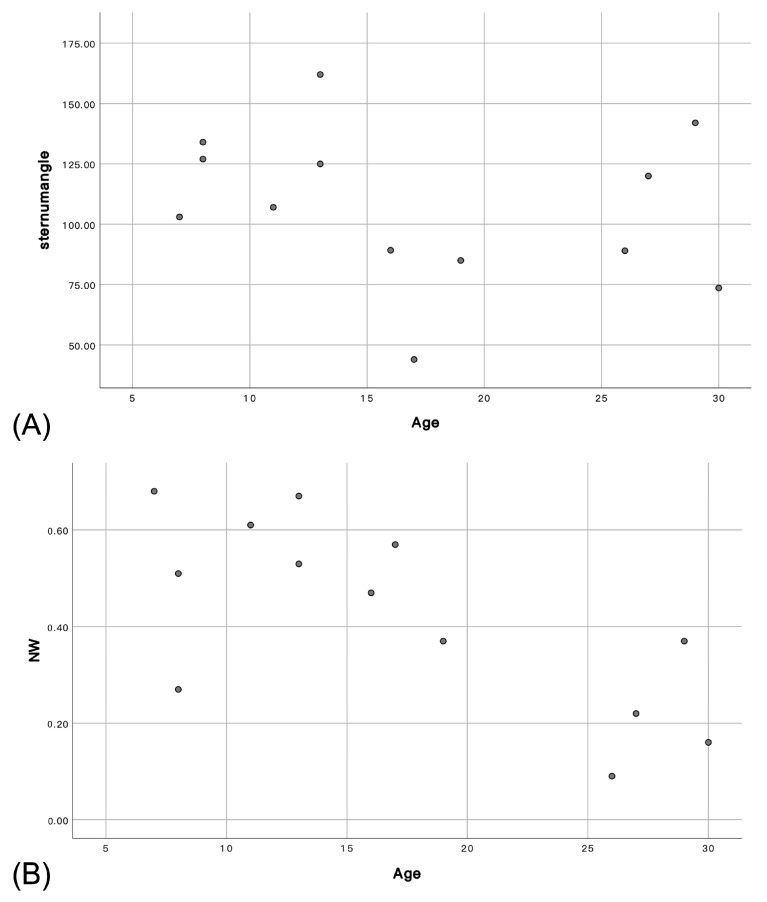
(**A**) Deformed chest and sternal angle had no correlation with age (*p* = 0.436). (**B**) NW ratio had a negative correlation with age (r = −0.708, *p* = 0.007).

**Figure 4 jpm-13-00494-f004:**
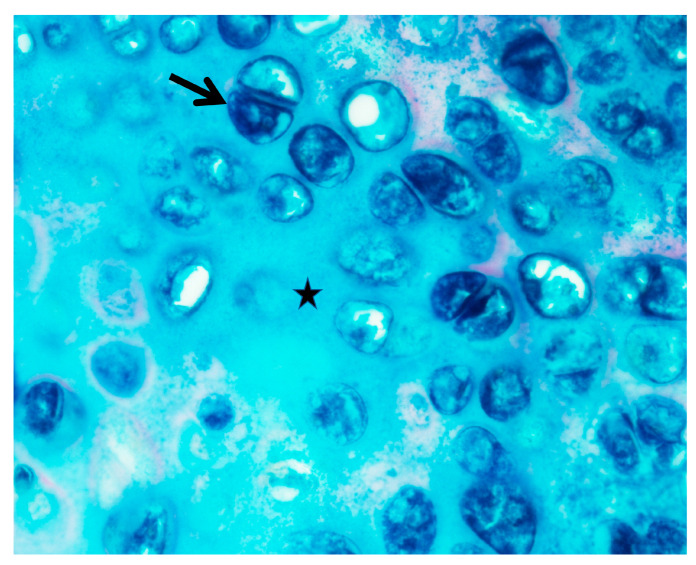
Pathological study with colloidal iron stain showing that GAGs were deposited in the cytoplasm of chondrocytes (arrow) and extra-cellular matrix (star) in the tracheal cartilage in case No. 8.

**Table 2 jpm-13-00494-t002:** Airway evaluation with FB and CT in patients with MPS type IVA.

No.			1	2	3	4	5	6	7	8	9	10	11	12	13	14	15
Sex			M	F	F	M	F	F	M	M	M	M	M	F	M	M	M
Age (years)			7	16	11	19	13	8	8	26	27	30	13	17	29	22	21
ERT			–	–	–	+ (3 years)	–	–	+ (1 year)	+ (5 months)	–	–	+ (10 years)	–	–	–	–
FB	Pharynx	Adenoids	2	2	2	2	2	3	0	1	2	2	4	4	2	1	2
		Tonsils	3	3	0	3	4	3	3	3	3	3	3	3	3	2	3
		Tongue	III	IV	IV	IV	IV	III	IV	IV	IV	IV	IIb	IV	IV	III	IV
		Prolapsed soft palate	–	+	+	+	–	+	+	+	+	+	–	+	+	+	+
	Larynx	2^0^ LM	–	+	+	++	+	++	++	++	++	++	–	++	++	+	++
		VC granulation	–	+	–	–	–	–	+	+	+	+	–	+	+	–	+
		Cricoid thickness	+	+	+	+	–	+	+	–	+	+		+		–	+
	Trachea	Stenosis	I	II	I	II	I	I	III	III	III	III				I	III
		Shape (upper/middle/lower)	C/C/C	U/D/T	U/D/D	T/U/T	C/U/C	U/U/C	T/W/U	B/D/T	W/T/T	B/T/T				C/C/C	W/U/U
		Deposit node	–	+	–	–	–	–	–	–	–	+				–	–
		Kinking	–	+	–	–	–	–	–	–	–	+				–	–
		TM with rigid lumen	–	–	–	+	–	–	+	+	+	+				–	–
	Bronchus	collapse	–	–	–	–	–	–	–	–	–	–				–	–
CT	Chest	Deformity	++	+++	++	+++	–	++	++	+++	+	+++	++	+++	+		
		Sternum angle (°)	103 ^·^	89.2	107	85	162	127	134	89	120	73.6	125	44	142		
	Trachea	Collapse	–	++	+	++	+	+	++	+++	+++	+++	–	+	++		
		NW ratio	68%	47%	61%	37%	53%	51%	27%	9%	22%	16%	67%	57%	37%		
		Torsion	–	–	+	+	–	–	+	-	+	+	–	+	+		
		Kinking	–	+	–	–	–	–	–	–	–	+	–	+	–		
		External compression	–	+ (I+A)	+ (I)	+ (I)	–	–	+ (I+A)	+ (I)	+ (I)	+ (I+A)	–	–	+ (I)		
		Framework damage	–	–	+	–	–	–	–	–	–	+	–	–	+		
Operation			–	T+A+LM	T+A	A	T+A+LM	–	A A T+LM	T-tube+T+A+LM	T-tube	T-tube +LM	–	–	–	–	–
Age at operation (years)			–	16	9	13	13	–	6 7 8	26	27	31	–	–	–	–	–
Pathology (Colloidal iron stain)			–	Not tested	Tonsils (+) Adenoids (+)	Adenoids (–)	Tonsils (–) Adenoids (–)	–	Tonsils (+) Adenoids (–)	Trachea (+) Arytenoid (–) Tonsil (+)	Deep neck soft tissue (–) Trachea (+)	Arytenoid (+) Cricoid (+)	–	–	–	–	–

FB, fiberoptic bronchoscopy; CT, computed tomography; 2^0^ LM, secondary laryngomalacia; VC, vocal cord; TM, tracheomalacia; NW ratio of narrowest to widest areas; I, innominate artery; I+A, innominate artery and aortic arch.

**Table 3 jpm-13-00494-t003:** Distribution of the grading of parameters of fiberoptic bronchoscopy, computed tomography, and surgery type.

Variables	N (%)
**Fiberoptic-bronchoscopy**	
**Pharynx (*N* = 15)**	
Adenoids	
0	1 (6.7)
1	2 (13.3)
2	9 (60.0)
3	1 (6.7)
4	2 (13.3)
Tonsils	
0	1 (6.7)
1	0 (0.0)
2	1 (6.7)
3	12 (80.0)
4	1 (6.7)
Tongue	
I	0 (0.0)
IIa	0 (0.0)
IIb	1 (6.7)
III	3 (20.0)
IV	11 (73.3)
Uvula in larynx (+)	12 (80.0)
**Larynx (*N* = 15)**	
2^0^ LM	
–	2 (13.3)
+	4 (26.7)
++	9 (60.0)
VC granulation (+)	8 (53.3)
Cricoid thickness (+)	10 (76.9)
**Trachea and Bronchus (*N* = 12)**	
Stenosis	
I	5 (41.7)
II	2 (16.7)
III	5 (41.7)
IV	0 (0)
Shape ^a^	
C	4 (33.3)
U	7 (58.3)
D	3 (25.0)
T	6 (50.0)
W	3 (25.0)
B	2 (16.7)
Deposit nodule (+)	2 (16.7)
Kinking (+)	2 (16.7)
TM with rigid lumen (+)	5 (41.7)
Bronchus collapse (+)	0 (0.0)
**Computed tomography**	
**Chest**	
Deformity	
–	1 (7.7)
+	2 (15.4)
++	5 (38.5)
+++	5 (38.5)
Sternal angle (°)	107.8/107.0
**Trachea**	
Stenosis	
–	2 (15.4)
+	3 (23.1)
++	5 (38.5)
+++	3 (23.1)
NW ratio (%)	42.5/47.0
Torsion (+)	8 (61.5)
Kinking (+)	3 (23.1)
External compression (+)	8 (61.5)
Framework damage (+)	3 (23.1)
**Surgery type and Pathology**	
**Surgery type (*N* = 15)**	
Tonsillectomy	5 (33.3)
Adenoidectomy	6 (40.0)
Laser supraglottoplasty	5 (33.3)
Tracheostomy with T-tube stenting	3 (20.0)
Pathology (Colloid iron stain)	
Tonsil	3/5 (60.0)
Adenoid	1/6 (16.7)
LM	1/5 (20.0)
Neck	0/3 (0.0)
Trachea	2/3 (66.7)
Cricoid	1/3 (33.3)

2^0^ LM, secondary laryngomalacia; VC, vocal cord; TM, tracheomalacia; NW ratio between narrowest and widest area. ^a^ Definition is shown in Table 1.

## Data Availability

All data are present within the article and can be further obtained from the corresponding author on reasonable request.

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
