# Peer review of "Endoscopic and Image Analysis of the Airway in Patients with Mucopolysaccharidosis Type IVA"

_jpm, 2023, doi:10.3390/jpm13030494_

Round 1

Reviewer 1 Report

Dear Authors,

 Thank you for the opportunity to review your precise paper. This paper described the evaluation about patients with MPS type IVA with obstructive airways via endoscopy and imaging modalities, and the effects of surgical interventions. 

  There has not been reported about the detail about respiratory problems so far, it will be useful and informative for clinician and so on.  

  On the other hand, I would like to ask authors some points as the following and request the author the minor revision.

 ãƒ»I think tracheal kinking was resulted from discordance of body height and size of inner organ. Is there any data indicating the relationship between body height and tracheal kinking? If you have a data about the body height, it would be great to add the information into the paper.

・Author mentioned ERT doesn’t the prevent the progression of tracheal problem,  I just wonder that early start of ERT will improve the dysfunction of airway or not. Could you describe more detail in the discussion?

・If you have a figure of pathophysiological findings, please include in to figure.

Author Response

Response to Reviewer 1 Comments

Dear reviewer,

Thank you for your time and opinion to my work. Your opinion and question provide me some important points that I missed before.

Point 1:

・I think tracheal kinking was resulted from discordance of body height and size of inner organ. Is there any data indicating the relationship between body height and tracheal kinking? If you have a data about the body height, it would be great to add the information into the paper.

 Response 1:

Yes, I do agree tracheal kinking results from discordance of body height and size of inner organ. I followed your suggestion and reviewed the basic data of individual cases about their body height and weight. However, since this is a retrospective study and the cases were collected for the past 20 years, it was difficult to comfirm the precise data of every case. Airway was usually evaluated by ENT doctors, but the basic physical data, such as body weight and height, was usually recorded by pediatricions . Four out of 15 patients visited the ENT clinic on a much different date (from 2 to 5 years) than the pediatric department’s clinic. These four cases were children and teenagers and were still under growing, so it is hard to tell that their body height changed or not when undergoing airway examination. Thus, if I list all the body height of these 15 cases, there may be some statical erro in some cases.

Point 2: Author mentioned ERT doesn’t the prevent the progression of tracheal problem,  I just wonder that early start of ERT will improve the dysfunction of airway or not. Could you describe more detail in the discussion?

Response 2:

I added some description in the section of ERT.

The limited impact of ERT on bony and cartilagous lesions in MPS IVA patients has been described detaily in the reference [3] (Kazuki Sawamoto et al. Mucopolysaccharidosis IVA Diagnosis, Treatment, and Management. Int J Mol Sci. 2020 Feb; 21(4): 1517.). The obstructive and restrictive airway in MPS type IVA were closely related to the devastating skeletal dysplasia and cartilaginous deformity. However, since ERT has limited impact on bone and cartilage, airway symptoms seem to have limited improvement even after long term ERT.

Point 3: If you have a figure of pathophysiological findings, please include in to figure.

Response 3:

I added Figure 4.

Figure 4. Pathological study with colloidal iron stain showed that GAGs were deposited in the cytoplasm of chondrocyte (arrow) and extra-cellular matrix (star) in tracheal cartilage in case No. 8.

Reviewer 2 Report

I was very pleased to review the manuscript „Endoscopic and Image Analysis of the Airway in Patients with Mucopolysaccharidosis type IVA“. The manuscript is well-written and comprehensive. Retrospective data collected from 15 MPS IVA patients from a single center nicely represent well-known airway issues in MPS IVA. I find that the figures are very illustrative and add significant value to this paper. I have only a few minor suggestions:

1.      In the abstract: NW ratio – please use the full term and the abbreviation in brackets

2.      In the introduction: In the second sentence the bones are mentioned twice unnecessarily  „... storage of GAGs in the connective tissues and bones throughout the body, particularly in the bones, ...“ I suggest changing it to „...storage of GAGs in the connective tissues, particularly in the bones, central and peripheral nervous system, liver, ..."

Regarding MPS classification you are mentioning MPS I H/S/H-S, but not MPS III A/B/C/D or MPS IV A/B. I suggest deleting H/H-S/S or adding subtypes for other diseases. You omitted to mention MPS IX (hyaluronidase deficiency).

Please consider citing the paper Akyol, M.U., Alden, T.D., Amartino, H. et al. Recommendations for the management of MPS IVA: systematic evidence- and consensus-based guidance. Orphanet J Rare Dis 14, 137 (2019), which among other disease complications addresses airway problems, follow-up and management.

Author Response

Response to Reviewer 2 Comments

Dear reviewer,

Thank you for your time and opinion to my work. Your opinion and question provide me some important points that I missed before.

Point 1: In the abstract: NW ratio – please use the full term and the abbreviation in brackets.

 Response 1:

I added the full term of ”narrowest-widest ratio” and the abbreviation (NW ratio) in brackets in the abstract.

Point 2: In the introduction: In the second sentence the bones are mentioned twice unnecessarily  „... storage of GAGs in the connective tissues and bones throughout the body, particularly in the bones, ...“ I suggest changing it to „...storage of GAGs in the connective tissues, particularly in the bones, central and peripheral nervous system, liver, ..."

Response 2:

I deleted the first repeated “bones” by your suggestion.

Point 3:  Regarding MPS classification you are mentioning MPS I H/S/H-S, but not MPS III A/B/C/D or MPS IV A/B. I suggest deleting H/H-S/S or adding subtypes for other diseases. You omitted to mention MPS IX (hyaluronidase deficiency).

Response 3:

I added the subtype of type III A/B/C/D, type IV A/B, and type IX (Natowicz syndrome) by your suggestion.

Point 4:  Please consider citing the paper Akyol, M.U., Alden, T.D., Amartino, H. et al. Recommendations for the management of MPS IVA: systematic evidence- and consensus-based guidance. Orphanet J Rare Dis 14, 137 (2019), which among other disease complications addresses airway problems, follow-up and management.

Response 4:

I cited the paper: “Akyol, M.U.; Alden, T.D.; Amartino, H.; Ashworth, J.; Belani, K.; Berger, K.I.; Borgo, A.; Braunlin, E.; Eto, Y.; Gold, J.I.; et al. Recommendations for the management of MPS IVA: systematic evidence- and consensus-based guidance. Orphanet J Rare Dis 2019, 14, 137, doi:10.1186/s13023-019-1074-9” as my 25th reference.